# Mycoplasma Co-Infection Is Associated with Cervical Cancer Risk

**DOI:** 10.3390/cancers12051093

**Published:** 2020-04-28

**Authors:** Cameron Klein, Kandali Samwel, Crispin Kahesa, Julius Mwaiselage, John T. West, Charles Wood, Peter C. Angeletti

**Affiliations:** 1Nebraska Center for Virology, School of Biological Sciences, University of Nebraska-Lincoln, Lincoln, NE 68588, USA; 2Ocean Road Cancer Institute, Dar es Salaam 3592, Tanzania; 3Nebraska Center for Virology, Department of Biochemistry, University of Nebraska-Lincoln, Lincoln, NE 68588, USA

**Keywords:** cervical cancer, HPV, HIV, Mycoplasma genitalium, Mycoplasma hominis, Ureaplasma, Lactobacillus iners, Sub-Saharan Africa, Tanzania

## Abstract

Tanzania faces one of the highest cervical cancer burdens in the world. Recent work has suggested that the bacterial family *Mycoplasmataceae* is associated with higher levels of human papillomavirus (HPV), human immunodeficiency virus (HIV), and pre-cancerous cervical lesions. *Mycoplasmataceae* infection in Tanzania is not well understood, especially when considering the differences between sexually transmitted species of *Mycoplasmataceae*. To establish the prevalence of common *Mycoplasmataceae* cervical infections and evaluate their relationship with risk factors for cervical cancer, 1160 Tanzanian women responded to an epidemiological questionnaire and were tested for HIV, HPV, cervical lesions, *Mycoplasma genitalium*, *Mycoplasma hominis*, *Ureaplasma* spp., and *Lactobacillus iners*. A subset of 134 women were used for 16s metagenomic sequencing of cervical DNA to establish the relative abundance of *Mycoplasmataceae* and *Lactobacillus* present. PCR detection of bacteria at the cervix found *Ureaplasma* spp. in 51.4% of women, *M. hominis* in 34%, *M. genitalium* in 2.3%, and *L. iners* in 75.6%. *M. hominis* and *M. genitalium* infection were significantly more prevalent among women with HPV and HIV. *M. hominis* prevalence was similar despite severity of cervical lesions; however, abundance of *M. hominis* increased significantly in women with cervical lesions. These results emphasize the importance of understanding the relationship between *M. hominis* and HPV-related cervical pathogenesis.

## 1. Introduction

Cervical cancer mortality is higher in Eastern Africa than in any other region of the world [1]. In Tanzania, cervical cancer is the most prevalent cancer in females [2]. Tanzania faces many issues which contribute to the burden of cervical cancer, including high human papillomavirus (HPV) prevalence, high human immunodeficiency virus (HIV) prevalence, low condom use, irregular preventative screening, and lack of full implementation of the pap smear. In Europe and the U.S., preventative screening for cervical cancer is usually done by HPV testing or checking for lesions in the cervical epithelium via a pap smear. In Tanzania however, cervical screening is mainly visual inspection with acetic acid (VIA), which is markedly less sensitive for early detection of cervical lesions than the pap smear and does not grade lesions by severity. Cervical lesions detected during screening are usually associated with HPV infection; however, recent studies have proposed that the cervical microbiome may be an important co-factor for the development of pre-cancerous and cancerous lesions [3]. Currently, it is not understood how, or which cervical microbiota contribute to cervical lesions, although in a previous study, we found that the bacterial family *Mycoplasmataceae* was the most significant differential cervical bacteria between women with normal cervical cytology and those with pre-cancerous lesions in Tanzania [4]. *Mycoplasmataceae* are the smallest known bacteria, in both physical and genomic size. During infection of the cervicovaginal epithelium, *Mycoplasmataceae* establish a persistent, intracellular infection which can lead to inflammatory-cytokine-mediated tissue injury. Although it is currently unknown if there is a mechanistic relationship between HPV and *Mycoplasmataceae*, the nature of *Mycoplasma* infection allows for direct interaction with HPV during co-infection of a single cell, and indirect interaction through cytokine responses.

*Mycoplasmataceae* is comprised of the genera *Mycoplasma* and *Ureaplasma*, which include several sexually transmitted species with global prevalence. Most notably, *M. hominis*, *M. genitalium*, and *U. urealyticum* are relatively common sexually transmitted infections (STIs) associated with cervical inflammation [3,5,6]. Among *Mycoplasma*, only *M. genitalium* is sometimes included in regular STI screening, although *M. hominis* is believed to have a similar pathogenesis. As a result, *M. hominis* has received significantly less study, and its relationship with HPV, HIV, and cervical lesions remains unclear. The prevalence of *M. genitalium, M. hominis,* and *U. urealyticum* in Tanzania has not previously been established in a large and diverse cohort, nor has it been considered alongside established risk factors for cervical dysplasia.

It has been suggested that high levels of cervicovaginal dysbiosis and transmission of *Mycoplasma* and other STIs in Eastern Africa is, in part, due to the commensal cervicovaginal bacteria in the region. Specifically, *L. iners* is the most prevalent cervicovaginal *Lactobacillus* in Eastern Africa, especially in HIV+ women, but has been shown to be less protective against cervicovaginal infection than other *Lactobacillus* [7]. Whether a cervical microbiome dominated by *L. iners* is conducive to infection and proliferation of *Mycoplasmataceae*, and the relationship between the bacteria and HPV pathogenesis, remains unclear. This study aims to establish the prevalence of common *Mycoplasmataceae* species in Tanzania and evaluate their relationship with *L. iners* and risk factors for cervical cancer, including HPV, HIV, and lifestyle factors.

## 2. Results

### 2.1. Cohort Demographics

DNA was successfully isolated from the cervical cytobrush samples of 1060 women. Complete data of cervical cytobrush DNA, pap-smear, VIA, HIV status, and epidemiological questionnaire response was available for 1002 women. Women with incomplete data were included in analyses where the missing data were not relevant. The cohort averaged 38.3 years of age, ranging from 18 to 73. A large majority (92.3%) of the women screened reported at least one previous pregnancy, and 84.1% were sexually active within the 3 months preceding sampling. A total of 67.4% of the cohort reported the use of at least one type of birth control, although it is unclear if they had recently used birth control at time of sampling. A total of 17.6% of the cohort had tested positive for HIV and these participants were on antiretroviral therapy at the time of sampling. Using a multiplex HPV genotyping PCR, we found that 46.1% of the cohort tested positive for at least one HPV genotype, and 38.2% of HPV positive women were coinfected with at least two genotypes.

There was a significant difference between the identification of cervical lesions between VIA and pap smear. Only 17% of women with pap smears graded HSIL had lesions identified by VIA. Additionally, although 88.9% of the cohort tested negative for cervical lesions by VIA, only 20.1% of the cohort was graded NILM by pap smear. The majority of women had pap smears exhibiting low-grade cervical dysplasia: ASCUS (24.8%) and LSIL (30.8%). More severe cervical dysplasia was apparent in 24.4% of women (14.9% ASC-H, 9.5% HSIL).

### 2.2. Mycoplasma Screen

Table 1 shows the breakdown of all data collected in this study and the variation of cervical cancer risk factors and *Mycoplasmataceae* prevalence within each group. PCR detection of *Mycoplasmataceae* at the cervix found a high prevalence of the bacterial family among Tanzanian women—66% of whom tested positive for at least one *Mycoplasmataceae*. *Ureaplasma* spp. was the most prevalent *Mycoplasmataceae*, detectable in 51.4% of the cohort, followed by *Mycoplasma hominis* in 34%, and *Mycoplasma genitalium* in only 2.3% of women. *Lactobacillus iners* was more prevalent than *Mycoplasmataceae*—detectable in 75.6% of women. Detection of any *Mycoplasmataceae* significantly increased the likelihood of detection of other *Mycoplasmataceae* species in that individual (Appendix AB–E). Women with *L. iners* also had higher prevalence of *Ureaplasma* spp. and *M. hominis* than woman without *L. iners*. Both *M. hominis* and *M. genitalium* were more common in women who reported previously having been diagnosed with an STI, though it is unclear if the STI was *Mycoplasma* related (Appendix AC,D). Mycoplasma was prevalent amongst all age groups.

### 2.3. Effects of HIV Infection

Being HIV+ increased the odds of detection of all *Mycoplasmataceae* and *L. iners*. *Mycoplasma hominis* and *genitalium* infections were especially prevalent among HIV+ women when compared to HIV− (odds ratio (OR) 4.8 and 4.2, respectively), while *Ureaplasma* spp. and *L. iners* were only slightly more common (OR 1.6 for both). This data supports previous research suggesting the HIV+ population acts as a reservoir for *M. hominis* infection [8]. Mycoplasma was still quite prevalent among HIV− women (49.5%, 27.6%, 1.5% prevalence for *Ureaplasma* spp., *M. hominis*, and *M. genitalium*, respectively). 

### 2.4. Effects of HPV Infection

Women infected with at least one HPV genotype were significantly more likely to have cervical dysplasia, especially high-grade lesions (OR 1.3773 for non-NILM, OR 2.7108 for HSIL). HPV+ women were also more likely to be infected with *M. hominis* (OR 2.1, *p* < 0.0001), while *M.* genitalium and Ureaplasma did not have a significant increase in prevalence associated with HPV (Figure 1). Commensal bacteria *L. iners*, was more likely to be present in HPV+ women (OR 2.0, *p* < 0.00001). Co-infection with two or more different HPV genotypes was associated with higher prevalence of *M. hominis* and *M. genitalium* than women infected by one HPV genotype. Multiple HPV infections were much more common amongst HIV+ women; however, this increase in *Mycoplasma* prevalence was also apparent in HIV− women with multiple HPV when compared to HIV−, single HPV women.

### 2.5. Effects of Cervical Cytology

*Mycoplasmataceae* were not significantly more or less prevalent among women with cervical dysplasia (Figure 1). Multivariate analysis of cervical cytology found that prevalence of HPV, number of pregnancies, use of injection-based birth control, and self-reporting of a previous STI varied significantly between cytology groups (Appendix AA). Only HPV prevalence had an obvious positive relationship with severity of cervical lesions, while having more than five pregnancies or using injection-based birth control were associated with increased odds of high-grade cervical lesions.

### 2.6. Effects of Other Factors

Sexual history was an important factor for detection of *Mycoplasmataceae* and *L. iners*. Women with three or more unique previous sex partners were significantly more likely to be infected with *M. hominis*, HPV, and HIV and were more likely to test HSIL. Prevalence of *M. hominis* and *L. iners* was significantly higher among women who had been sexually active during the 3 months prior to sampling (Appendix AC,E). Self-reported condom use was very low, especially for HIV− women (2.6%), contributing to increased transmission of Mycoplasma among sexually active women. Aging was associated with a significant decrease (*p* = 0.0004) in *L. iners* prevalence, decreasing from 83.7% in women 18–29 to 59.4% in women 50+. Age did not appear to be related with a shift in *Mycoplasmataceae prevalence*, although women aged 50+ did have somewhat lower prevalence of *M. hominis* and *M. genitalium*, possibly related to menopause or decreased sexual activity. *L. iners* prevalence also decreased in women with three or more previous pregnancies; however, this may have been influenced by a higher average age among high-gravidity women. 

### 2.7. Relative Abundance

A subset of 104 cervical samples was analyzed via 16s metagenomic sequencing to establish the relative abundance of *Mycoplasmataceae* and *L. iners* present. Each sample was rarefied to an even depth of 1000 reads. After rarefication, women with more than 5 reads from *Ureaplasma* spp., *M. hominis*, *M. genitalium*, *or L. iners* were considered positive for that bacteria. The prevalence of each bacteria was similar to results from PCR screening, although no *M. genitalium* reads were present among the subset of samples tested. By using the number of reads generated, we were able to determine the relative abundance of each bacteria in each woman’s cervical microbiome. Using this, we estimated the relative abundance of the screened bacteria within cervical cytology groups by adjusting the prevalence of a bacteria by the average relative abundance of that bacteria in positive samples of each cytology grade. When looking at the relative abundance of Mycoplasmataceae in women with cervical dysplasia, it becomes apparent that a significantly larger portion of the cervical microbiota is *M. hominis* (Figure 2). *M. hominis* is the only Mycoplasmataceae which increases linearly with the development of more severe cervical lesions. *Ureaplasma* spp. were most abundant among HSIL women; however, LSIL had a lower abundance than NILM women. *L. iners* was least abundant among HSIL women, but significantly more abundant in LSIL than NILM women. *Lactobacillus crispatus* is considered to be the most protective cervicovaginal microbe. Although we did not PCR screen for *L.* crispatus, no women with *L. crispatus* reads by 16s had *M. hominis*, suggesting *L. crispatus* protects against Mycoplasma infection, while *L. iners* does not.

## 3. Discussion

In this study, we found that cervical Mycoplasma infection is prevalent among Tanzanian women. Even though *M. genitalium* is more often screened for as a cervicovaginal infection, we found that *M. hominis* and *Ureaplasma* spp. were significantly more common in Tanzania. Similar results have been found in *M. hominis* and *M. genitalium* screens from other sub-Saharan African countries [9,10,11,12]. The primers we used to detect *Ureaplasma* spp. included both *U. urealyticum* and *U. parvum*. *U. urealyticum* is known to be a cervicovaginal pathogen; however, *U. parvum* is sometimes commensal in the uterus. Because we took our samples partially from the endocervix, it is likely *U. parvum* originating from the cervical opening may also have been detected. For this reason, we did not consider *Ureaplasma* spp. as a non-commensal infection and focus on the importance of *M. hominis* as a common, poorly understood cervical infection in Tanzania. 

Women who reported having had an STI were more likely to have an *M. hominis* or *M. genitalium* infection; however, most women with such an infection did not report any history of STIs. This indicates that most *M. hominis* and *M. genitalium* infections are asymptomatic, and thus go untreated. Currently, it is unclear how long a *Mycoplasma* infection of the cervix can persist while untreated. We detected higher prevalence of *M. hominis* among sexually active women, even those with a single long-term partner, suggesting sex may be important for persistence of *M. hominis* infection. Despite similar prevalence, significantly higher abundance of *M. hominis* in the presence of cervical lesions, especially high-grade cervical lesions, suggests that proliferation of *M. hominis* and development of cervical lesions have some form of mechanistic relationship. It is possible proliferation of *M. hominis* drives the formation of cervical lesions, or that cervical lesions create a microenvironment that favors proliferation of *M. hominis*. Longitudinal sampling of *M. hominis* abundance and cervical cytology would help to clarify this relationship. *M. hominis* may also contribute to HPV-driven cervical lesion formation by increasing persistence of the pathogens during co-infection. We found that prevalence of *M. hominis* was significantly higher among HPV+ women, which could result from prolonged persistence increasing the likelihood of sampling an infection. This idea is supported by previous studies which have identified cervical pathogens, including *Mycoplasma*, as cofactors in the persistence of HPV infection [13,14,15,16]. The intracellular nature of *Mycoplasma* infection is particularly interesting when considering its relationship with HPV. Intracellular bacterial infections may directly interact with HPV replication in epithelial cells, while also contributing to the epithelium’s immune microenvironment by influencing cytokine expression. The data presented here highlights the need for further research into *M. hominis* prevalence and pathogenesis, especially related to HPV, HIV, and cervical cancer.

Our data support previous research suggesting *L. iners* is an especially common commensal cervical bacteria in Sub-Saharan African countries. Increased prevalence of commensal *L. iners* among HIV+, HPV+, and *Mycoplasmataceae*+ women suggests that *L. iners* does not protect the cervix from infection, as other Lactobacillus species are believed to do. This is further evidenced by our 16s data, where co-detection of *M. hominis* and *L. iners* was common, but *M. hominis* and *L. crispatus* were never detected together.

This study highlights the need to account for significant regional differences in cervicovaginal microbiota, especially Mycoplasma. The high prevalence of M. hominis and its association with risk factors for cervical cancer (HPV, HIV, and cervical lesions) demonstrates the importance of better understanding *M. hominis* pathogenesis. Our results suggest screening for *Mycoplasma* is especially important in Tanzania, particularly among women at high risk for cervical cancer. Establishing a screening and treatment protocol to address the prevalence of asymptomatic *Mycoplasma* infection could reduce transmission of HPV and HIV by reducing susceptibility to infection, and potentially prevent progression of cervical lesions. Long-term, longitudinal studies are needed to clarify whether *Mycoplasma* becomes abundant at the cervix preceding or following the development of lesions, which would help to clarify if *Mycoplasma* is driving formation of cervical lesions or benefitting from the microenvironment associated with lesions.

## 4. Materials and Methods 

### 4.1. Participants and Ethical Precautions

This study reports findings derived from an ongoing cross-sectional cohort study analyzing demographics of HPV and cervical cancer in HIV-positive and -negative women from rural and urban Tanzania. Between March 2015 and February 2017, female patients undergoing cervical cancer screening were approached for enrollment in the study. Those who were pregnant, menstruating, under 18, reported being sick in the past 30 days, or had a preexisting, non-HIV, immunologic defect were excluded from the study. Disease histories and physical examinations were used to rule out any clinical symptoms or visible signs for these conditions. Samples were collected at three sites in Tanzania: Ocean Road Cancer Institute (ORCI) in Dar es Salaam and rural clinics in Chalinze and Bagamoyo. After collection of cervical samples and demographic data, samples from 1060 women were screened for *Mycoplasma* species and *Lactobacillus iners*. A subset of 132 women were also used for 16s metagenomic sequencing.

### 4.2. Demographic Data Collection

This study was approved for human subjects work by the University of Nebraska-Lincoln Institutional Review Board (IRB) under protocol ID: 14709. All study participants gave informed consent and were evaluated by study clinicians. A set of pretested, standardized questionnaires was used to gather demographic data. All personal identifiers were removed from samples to ensure patient confidentiality. With the permission of the patients, medical history was retrospectively retrieved from hospital medical records. More than 30 variables were identified and assessed in the questionnaire, including time since last sexual intercourse, number of sexual partners, number of pregnancies, use and type of birth control, and self-reported history of STI infections.

### 4.3. Specimen Collection, HIV and Pap Tests

Blood samples were collected via venipuncture into acid-citrate-dextrose tubes (Streck, Omaha, NE, USA) and processed using centrifugation at the on-site study laboratory within 6 h of being drawn. The separated plasma was tested at the ORCI, as part of standard of care, using Standard Diagnostics HIV-1/2 3.0 detection kit (Chembio, Medford, NY, USA). Cervical cytobrush samples and pap smears were collected from the cervical transformation zone of all patients. Pap smears were examined by at least three trained cytologists and classified according to the pap classification protocol: negative for intraepithelial lesion or malignancy (NILM); atypical squamous cells of undetermined significance (ASC-US); low-grade squamous intraepithelial lesions (LSIL); atypical squamous cells but cannot exclude high-grade lesions (ASC-H); high-grade squamous intraepithelial lesions (HSIL). Cervical cytobrush specimens were placed in lysis buffer (Qiagen, Redwood City, CA, USA) and then shipped to the Nebraska Center for Virology at the University of Nebraska-Lincoln (UNL) for processing.

### 4.4. DNA Isolation

Cervical cytobrush samples were vortexed and separated from the brush with lysis buffer. DNA was extracted from the lysis buffer using the DNeasy Tissue extraction kit (Qiagen, Redwood City, CA, USA) according to the manufacturer’s protocol. The DNA concentration was determined by UV spectrophotometer (Thermofisher, Waltham, MA, USA) at 260/280 nm.

### 4.5. HPV Genotyping

To determine HPV status, DNA samples were genotyped for HR-HPVs (types 16, 18, 30, 31, 33, 35, 39, 45, 51, 52, 56, 58, 59, and 66) and LR-HPVs (types 6 and 11) using a low-cost multiplex PCR assay [17].

### 4.6. Mycoplasmataceae and L. iners Screen

A multiplex PCR targeting *M. genitalium*, *M. hominis*, and *Ureaplasma* spp. was adapted from Stellrecht et al. [18], with the addition of primers targeting *L. iners* established in [19]. Primers were mixed with sample DNA and Qiagen Multiplex PCR Master Mix according to the manufacturer’s protocol.

The PCRs were performed in a final volume of 25 μL. The cycling conditions were as follows: an initial denaturation of 95 °C for 15 min, followed by 35 cycles, with 1 cycle consisting of denaturation at 94 °C for 15  s, and annealing and extension at 60 °C for 1 min, then a final elongation of 72 °C for 5 min. After amplification, DNA samples were run in 0.5% agarose gels containing Ethidium Bromide at 95 volts for 1 h. Gels were then imaged using a Bio Rad ChemiDoc MP Imaging System (Biorad, Hercules, CA, USA) to visualize bands.

### 4.7. Statistical Analyses

Multivariate analysis of variance (MANOVA) using one variable selected as fixed versus the other remaining dependent variables collected (*Ureaplasma* spp., *M. hominis*, *M. genitalium*, *L. iners*, HPV, HIV, Age, time since last sexual activity, number of sex partners, number of pregnancies, self-reporting of STI infection, use of birth control, and type of birth control used) was used to identify significant differences between women with different cervical cytology, HIV, or HPV status. The birth control types considered were pills, injections, condoms, implants, loop, and natural. Odds ratios were calculated to identify groups with significantly increased odds of HPV, HIV, or *Mycoplasmataceae*. A *p* value of 0.05 was the maximum considered to be significant throughout the study.

### 4.8. 16S rRNA Library Preparation, and Sequencing of the V4 Region

DNA samples were used for tag sequencing of the V4 hypervariable region of the 16S rRNA gene. A 250-bp section of the V4 region was amplified using universal primers described in reference 40. The PCRs were performed in 25 μL. The cycling conditions were as follows: an initial denaturation of 98 °C for 3 min, followed by 25 cycles, with 1 cycle consisting of denaturation at 98 °C for 30  s, annealing at 55 °C for 30  s, and extension at 68 °C for 45  s, and then a final elongation of 68 °C for 4 min. Following amplification, PCR products were analyzed on a 2% agarose gel to confirm correct product size. Normalized amplicons (1 to 2 ng/μL) from 144 samples were pooled together using an epMotion M5073 liquid handler (Eppendorf AG, Hamburg, Germany). Pooled libraries were sequenced using the Illumina MiSeq platform using the dual-index sequencing strategy outlined by Kozich et al. [20].

### 4.9. 16S Data Processing and Bacterial Community Analysis. 

The sequencing data obtained from the sequencer were subsequently analyzed using the Illumina MiSeq data analysis pipeline developed by the Fernando lab (described in detail at https://github.com/FernandoLab). Briefly, initial quality filtering was carried out to remove sequences that had ambiguous bases, incorrect lengths, and inaccurate assemblies. Subsequently, the quality-filtered reads were run through the UPARSE pipeline (http://www.drive5.com/uparse/) and subjected to chimera filtering and OTU clustering (at a similarity threshold of 97%), followed by the generation of an OTU table. Taxonomy was assigned to the OTUs using the assign_taxonomy.py command available in QIIME using the Greengenes database (May 2013). The OTU table was rarefied across samples to the lowest sample depth (1000 reads) using QIIME based on the Mersenne Twister pseudorandom number generator. All statistical analyses were performed with samples at an even depth. 

### 4.10. Ethics Statement

All human subject protocols were approved by safety committees at the Ocean Road Cancer Institute (ORCI) and UNL in accordance with the Helsinki Declaration. Participation by patients was entirely voluntary, and written patient consent was required for inclusion in the study.

## 5. Conclusions

In this study, we analyzed the cervical microbiota of 1160 Tanzanian women. We found that the bacterial family of *Mycoplasmataceae* was detected at the cervix at 66% of women tested. Among the *Mycoplasmataceae* family, *Ureaplasma* spp. was the most common, at about 51.4% of the cohort. *Mycoplasma hominis* was found in 34% of the cohort, while *Mycoplasma genitalium* was found in only 2.3% of the cohort. We found that HIV+ women had increased odds for all *Mycoplasmataceae* and *L. iners*. *Mycoplasma hominis* and *genitalium* infections were especially prevalent among HIV+ women compared HIV−. Those women who were HPV+ were more likely to have *M. hominis* (OR 2.1, *p* < 0.0001). Co-infection with multiple HPV genotypes was associated with higher a prevalence for *M. hominis* and *M. genitalium*. Multiple HPV genotype infections were more common amongst HIV+ women (Figure 1).

The prevalence of *Mycoplasmataceae* was not associated with cervical dysplasia. However, we found that the abundance of *Mycoplasmataceae* bacteria, such as *M. hominis,* increased with more severe cervical lesions, LSIL and HSIL. *Ureaplasma* spp. abundance was also associated with HSIL women (Figure 2). These data suggest that the abundance of *M. hominis* and *M. Ureaplasma* are associated with cervical dysplasia. We suggest that *M. hominis* infection of the cervix could lead to chronic inflammation, which favors HPV-related neoplasias.

## Figures and Tables

**Figure 1 cancers-12-01093-f001:**
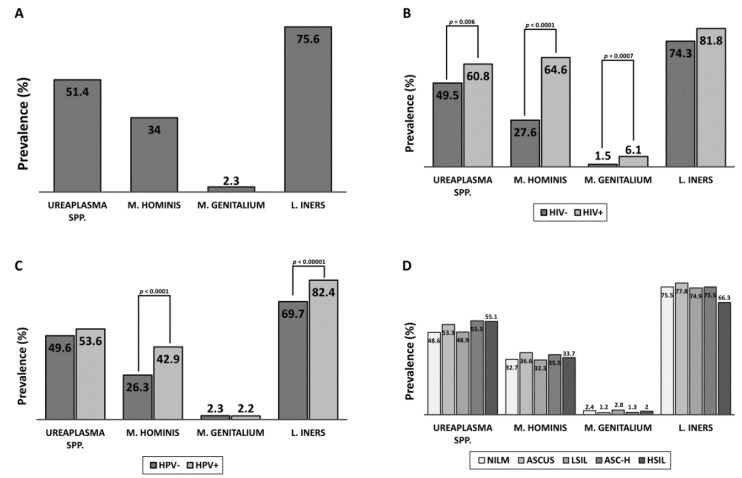
Relationship between *Mycoplasmataceae* abundance and cervical cancer risk factors. (**A**) Overall prevalence of the screened *Mycoplasmataceae* and *Lactobacillus* species in the cohort; (**B**) Comparison of prevalence between HIV+ and HIV− women; (**C**) Comparison of prevalence between HPV+ and HPV− women; (**D**) Comparison of prevalence between women based on cervical cytology.

**Figure 2 cancers-12-01093-f002:**
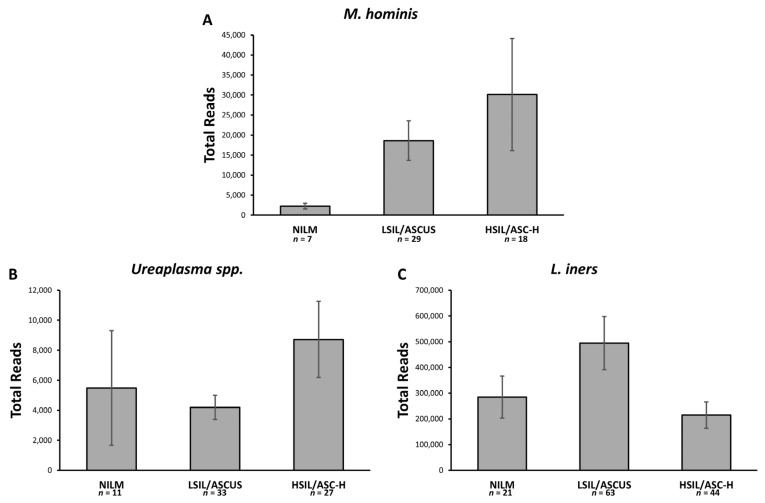
Relative abundance of *Mycoplasma* based on cervical cytology. Abundance among infected is the mean of positive 16s samples (*n*) adjusted by prevalence determined by PCR screen. Error bars represent standard error of the mean. (**A**) Expected number of *M. hominis* 16s DNA reads for 100 Tanzanian women of varying cervical cytology; (**B**) Expected number of *Ureaplasma* spp. 16s DNA reads for 100 Tanzanian women of varying cervical cytology; (**C**) Expected number of *L. iners* 16s DNA reads for 100 Tanzanian women of varying cervical cytology.

**Table 1 cancers-12-01093-t001:** Prevalence of Mycoplasma, HPV, HIV, and epidemiological factors.

Group	*n*	*Ureaplasma* spp.	*M. hominis*	*M. genitalium*	*L. iners*	HPV+	HIV+	NILM	LSIL^+^	HSIL^+^	Age
**Total**	1060	51.4	34	2.3	75.6	46.1	17.6	20.1	55.5	24.4	38.3
**HPV**											
**HPV+**	489	53.6	**42.9 ***	2.2	**82.4 ***	**--**	**24.8 ***	17.3	52.6	**30.1 ***	36.9
**HPV−**	571	49.6	**26.3 ***	2.3	**69.7 ***	**--**	**11.4 ***	22.4	58.1	**19.5 ***	39.5
**1 HPV**	302	55.3	38.1	**1 ***	79.8	--	19.4	20.7	51.7	27.6	37.2
**2+ HPV**	187	50.8	**50.8 ***	4.3	**86.6 ***	--	**33.5 ***	**11.9 ***	54	**34.1 ***	36.5
**HIV**											
**HIV+**	181	**60.8 ***	**64.6 ***	**6.1 ***	**81.8 ***	**65.2 ***	--	16.9	60.1	23	39.2
**HIV−**	847	49.5	**27.6 ***	1.5	74.3	**42.3 ***	--	20.8	54.6	24.6	38.1
**Cytology**											
**NILM**	208	48.6	32.7	2.4	75.5	39.9	14.9	--	--	--	38.4
**ASCUS**	257	53.3	36.6	1.2	77.8	**34.6 ***	19.5	--	--	--	37.6
**LSIL**	319	48.9	32.3	2.8	74.9	**35.7 ***	18.8	--	--	--	38.8
**ASC-H**	155	55.5	35.5	1.3	75.5	41.3	14.8	--	--	--	37.6
**HSIL**	98	55.1	33.7	2	66.3	50	19.8	--	--	--	39.5
**VIA**											
**Normal**	873	51.1	33.7	2.3	75.6	44.3	**14.7 ***	21	55.7	23.3	38
**Lesions**	109	49.5	35.8	2.8	79.8	**63.3 ***	**33 ***	15.7	55.6	28.7	37.1
**Age**											
**18–29**	209	54.5	33.5	2.4	**83.7 ***	52.6	**10.6 ***	21.4	53.4	25.2	25.8
**30–39**	374	50.3	37.7	2.9	76.7	50	20.9	18.1	55.5	26.4	34.5
**40–49**	321	51.1	33.6	2.2	74.8	41.7	19.2	21.6	58.1	20.3	44.1
**50+**	133	51.9	27.1	0.8	**59.4 ***	**36.1 ***	16	21.4	53.4	25.2	54.8
**Last Sex**											
**<3 months**	876	52.9	35.3	2.5	77.4	47.1	15.7	19.6	56.7	23.7	37.4
**4–12 months**	81	48.1	32.1	1.2	65.4	38.3	**29.1 ***	17.5	50	32.5	39.6
**>2 months**	85	42.4	**23.5 ***	1.2	**64.7 ***	44.7	26.2	28.2	50.6	21.2	46.6
**Sex Partners**											
**1–2**	455	47.9	**25.5 ***	1.5	**71.4 ***	**38.5 ***	**8.9 ***	20	59.1	20.9	38.6
**3–5**	465	54.8	**40.2 ***	2.8	77.4	**52.9 ***	**21.4 ***	19.2	54.3	26.5	37.7
**>5**	102	52	**44.1 ***	2.9	82.4	50	**36.6 ***	26.2	47.5	26.3	39.3
**Pregnancies**											
**0**	80	47.5	32.5	2.5	76.3	52.5	**8.9 ***	20.3	58.2	21.5	32.5
**1–2**	319	53.9	36.4	2.8	**82.8 ***	50.5	19.2	21	57.1	21.9	32.9
**3–5**	492	51	33.3	2	72.8	45.3	19.3	21.1	55.7	23.2	39.8
**>5**	151	51	32.5	2	68.2	**37.1 ***	13.2	15.4	51	**33.6 ***	47.7
**Birth Control**											
**No**	340	49.7	33.8	3.2	73.2	47.1	21.4	20.7	56.8	22.5	38.3
**Yes**	702	52.6	34.2	1.9	76.5	45.9	15.8	19.9	55.1	25	38.3
**Birth Control Type**											
**Pills**	370	51.6	34.6	1.4	75.1	42.7	17	19	56.6	24.4	40.7
**Injection**	416	54.6	35.8	2.6	78.1	48.8	15.9	20	**50 ***	**30 ***	37.4
**Condom**	44	43.2	40.9	4.5	81.8	**61.4 ***	**47.6 ***	27.3	50	22.7	35.9
**Implant**	124	57.3	37.9	1.6	78.2	42.7	12.9	19.1	**64.2 ***	**16.7 ***	33.9
**Loop**	70	47.1	24.3	2.9	67.1	42.9	**2.9 ***	20.6	47.1	32.3	43.8
**Natural**	12	41.7	33.3	0	**91.7 ***	41.7	16.7	25	58.3	16.6	40.1
**STI Self-Report**											
**No recent**	939	51.8	33.2	1.9	75.5	45.4	**15.1 ***	19.7	57	23.3	38.2
**Yes recent**	73	53.4	38.4	4.1	75.3	50.7	**37.5 ***	27.4	**42.5 ***	30.1	39
**PCR Detection**											
***Ureaplasma* spp.**	545	--	38	2.9	**80.6 ***	48.1	20.8	18.9	54.9	26.2	38.1
***M. hominis***	360	**57.5 ***	--	3.9	**83.6 ***	**58.3 ***	**33.3 ***	19.3	55.8	24.9	37.7
***M. genitalium***	24	66.7	**58.3 ***	--	83.3	45.8	**45.8 ***	23.8	57.1	19.1	35.5
***L. iners***	801	54.8	**37.6 ***	2.5	--	**50.3 ***	19	20.2	56.4	23.4	37.4

Values are listed as percentage of women positive for the condition labeled in each column. The cohort is broken down into sub-groups in each row, depending on results from testing or survey. A one-proportion *Z*-test was used to identify prevalence in subgroups that differ significantly from the cohort average. Values were considered significant when *p* < 0.05 and are bolded and labeled with a ‘*****’. The column ‘LSIL^+^’ includes LSIL and ASCUS pap smear results for ease of interpretation. Similarly, the column ‘HSIL^+^’ includes HSIL and ASC-H pap smear results. Abbreviations: HPV (Human Papillomavirus), HIV (Human Immunodeficiency Virus), NILM (Normal for Intraepithelial Lesion or Malignancy), LSIL (Low-Grade Squamous Intraepithelial Lesion), ASCUS (Atypical Squamous Cells of Unknown Significance), ASC-H (Atypical Squamous Cells-Cannot Rule Out High Grade), HSIL (High-Grade Squamous Intraepithelial Lesion).

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
