# Peer review of "Mycoplasma Co-Infection Is Associated with Cervical Cancer Risk"

_cancers, 2020, doi:10.3390/cancers12051093_

Round 1

Reviewer 1 Report

This manuscript describes an attempt to assess the extent to which co-infection or colonization with Mycoplasma species influences risk for cervical cancer.  Particular strengths of this study is the large cohort size and associated demographic data as well as a coincident 16S metagenomic study of a subset of participants. However, aspects of the presentation do not convey the results of their analysis and should be revised before acceptance for publication.

Table 1 is impossible to interpret in the format offered for review.  The authors may not be aware that the color coding of results described was not preserved and without p values it just becomes a list of numbers.

Is the title overstating the conclusions of the paper? Mycoplasmas were not significantly more or less prevalent in women with cervical dysplasia overall.  

The authors note that prior history of STI and women with >3 sex partners were more likely to test HSIL.  Yet the authors do not report whether any of the cohort had other STI infections.  While N. gonorrhoeae is not thought to be associated with cervical cancer, a preponderance of studies has implicated C. trachomatis.

The 16S read count data presented in Figure 2 is graphed without stats. Differential abundance analysis of microbial communities is under active development, to address the nature and relative sparsity of data so more information regarding the raw data here would be appropriate.  How were the groups for analysis balanced? How many participants fall into each of the cytological categories?  What controls did they use to ensure that they had adequately sampled the entire cervicovaginal community?  Where is the rest of the 16S data, will that be in a different manuscript? 

Author Response

Reviewer 1:

>Table 1 is impossible to interpret in the format offered for review.  The authors may not be aware that the color coding of results described was not preserved and without p values it just becomes a list of numbers.

The missing colors have been replaced with a one-proportion Z-test which identifies values in sub-groups that differ significantly from the cohort average. The legend has been updated to reflect this.

>Is the title overstating the conclusions of the paper? Mycoplasmas were not significantly more or less prevalent in women with cervical dysplasia overall.

Mycoplasma (especially hominis) are associated with higher prevalence among women with several risk factors of cervical cancer here (HPV, HIV, numerous sexual partners). This is the risk the title is referring to, furthered by the abundance of mycoplasma in cervical lesions.

>The authors note that prior history of STI and women with >3 sex partners were more likely to test HSIL.  Yet the authors do not report whether any of the cohort had other STI infections.  While N. gonorrhoeae is not thought to be associated with cervical cancer, a preponderance of studies has implicated C. trachomatis.

STI history was self-reported and therefore, the type of STI could not be confirmed.  Women with obvious current STIs (besides those tested in the study) were excluded from the study.

> The 16S read count data presented in Figure 2 is graphed without stats.

We added error bars and n values to figure 2 and adjusted legend.

>Differential abundance analysis of microbial communities is under active development, to address the nature and relative sparsity of data so more information regarding the raw data here would be appropriate.  How were the groups for analysis balanced? How many participants fall into each of the cytological categories?  What controls did they use to ensure that they had adequately sampled the entire cervicovaginal community?  Where is the rest of the 16S data, will that be in a different manuscript?

This information is available in our published study which focuses on the 16s data: DOI: 10.1128/mBio.02785-18.

Reviewer 2 Report

Cervical cancer is the second common cancer and the third leading cause of cancer deaths among women in less developed countries. It has been indicated that changes in vaginal microbiome play an important role in the occurrence and development of cervical cancer. It has been indicated that changes in vaginal microbiome play an important role in the occurrence and development of cervical cancer. In this regard, human papillomavirus (HPV) has been known as a potential cause of cervical cancer. Although HPV infection plays an important role in the pathogenesis of cervical cancer, studies indicated that it is not sufficient by itself, and other factors such as cervix microbial flora are likely associated with the development of the disease. It has been seen that Lactobacillus iners, generally found in the vagina of healthy women, plays an important role in vaginal microbiome and increase bacterial vaginosis especially Mycoplasma hominis. Although the exact relationship between dysbiosis (dysbacteriosis) and the development of CIN remains unknown, it has been recommended that dysbiosis damages epithelial cell and lets HPV's enters into the basal epithelial cells which results in prolonged virus life, elongated infection and inflammation, and dysplasia, and consequently carcinogenesis (S. Pourmollaei, et al.2020). This study represents an interesting experience from a less developed country and found that M. Hominis was correlated to cervical lesions.

I have some small questions :

introduction

1) line 35 modification in Europe screening is done with HPV test

2) line 37 it is written that colposcopy is not able to make a selection on the severity of the lesions. I believe this is not appropriate. I ask the authors to modify. In results 1) line 81 the VIA is not a level I exam so it cannot be compared to the pap test I would remove this comparison 2) the legend of table 2 must be explained much better it is difficult to understand including the colors that are not seen 3) Figure 3 is very grainy and can be redone more clearly

Author Response

Reviewer 2:

>1) line 35 modification in Europe screening is done with HPV test

We reworded the sentence: “In Europe and the U.S. preventative screening for cervical cancer is usually done by HPV testing or checking for lesions in the cervical epithelium via a pap smear.”

>2) line 37 it is written that colposcopy is not able to make a selection on the severity of the lesions. I believe this is not appropriate. I ask the authors to modify.

The standard of care in Tanzania is not colposcopy, but VIA.  The observation that we made was that VIA is not as sensitive as pap smear detection of cervical abnormalities. We modified the text as follows

“In Tanzania however, cervical screening is mainly visual inspection with acetic acid (VIA), which is markedly less sensitive for early detection of cervical lesions than the pap smear and does not assess cytological abnormalities.”

>1) line 81 the VIA is not a level I exam so it cannot be compared to the pap test I would remove this comparison

VIA is used in Tanzania as the primary diagnostic method, instead of pap smear.  This is the reason for the comparison of VIA and pap smear.  It’s worth noting that the same comparisons have been done in numerous paper, particularly in regard to cervical cancer screening in low to middle income countries.

>2) the legend of table 2 must be explained much better it is difficult to understand including the colors that are not seen.

Missing colors have been replaced with a one-proportion Z-test which identifies values in sub-groups that differ significantly from the cohort average. The legend has been updated to reflect this.

>3) Figure 2 is very grainy and can be redone more clearly

We have inserted a new version of the figure with all titles italicized, and the y-axis font enlarged slightly. I also changed the settings to increase resolution of the final figure.

Reviewer 3 Report

This a relatively novel study in cervical cancer research. Peter Angeletti et. al suggested that the infection of Mycoplasma may be also related to cervical cancer development. Our immune system can automatically clear most HPV infections, but some HPV infections can persist. Pre-infecting or co-infecting with other pathogens, such as mycoplasma and chlamydia, may help HPV to avoid immune surveillance. Therefore, larger-scale pathogen screening and epidemiological investigations are needed. Besides, co-infection models, including in vitro and in vivo models, need to be established to answer the mechanic questions.

Author Response

Reviewer 3: No comments to address.
